# The spatiotemporal evolution of flight-coupled wind field for a four-rotor plant protection unmanned aerial vehicle

**Fengbo Yang** ⬤*, **Zhiwei Li, Guangyao Zhang, Hongping Zhou**

College of Mechanical and Electronic Engineering, Nanjing Forestry University, Nanjing, China

* yfb@njfu.edu.cn

## Abstract

The four-rotor plant protection Unmanned Aerial Vehicle (UAV) is an important piece of equipment for the efficient plant protection field. However, the flight-coupled wind field is unclear, which has become the bottleneck factor limiting the improvement of the spray quality. When the multi-rotor plant protection UAV flew and sprayed, the liquid droplets were accelerated, the stems and branches were shaken, and the leaves were turned over in the disturbance area of the spiral backward flight-coupled wind field. The flight-coupled wind field, liquid droplets and crop canopy were closely related. Only when the flight-coupled wind field was analyzed could the interaction mechanism among the flight-coupled wind field, liquid droplets, and crop canopy be effectively studied, then the flight and spray scheme could be reasonably formulated. By combining the Re-Normalization Group (RNG) $k$-$\varepsilon$ turbulence model, compressible Reynolds Averaged Navier-Stokes (RANS) equation, the dynamic mesh based on the spring smoothing and layering method, pressure-velocity coupling algorithm, a computational fluid dynamics (CFD) model of the flight-coupled wind field for the four-rotor plant protection UAV was established, the dynamic evolution law of the flight-coupled wind field in the spatial and temporal dimensions was also discussed in this paper. Numerical simulations were carried out for flight-coupled wind fields in two working conditions, analysis showed that the maximum relative error between the simulated and measured values of $Z_b$-direction ($Z$ direction in the absolute coordinate system) velocity was less than 12.7% when the flight-coupled wind field was stable. When switching from hover to flight state, the downwash wind field experienced a lateral interruption and developed into the stable flight-coupled wind field after flying for 0.696 s, and the velocity distribution diagram of the cross section evolved from four rings to four horseshoe vortices. When the flight-coupled wind field evolved to stabilize, the dense atmosphere reduced the absolute values of the four $Z_b$-direction velocity peaks at observation line 1 from left to right along the flight direction by 4.5%, 4.2%, 9.0%, and 26.1% respectively. The angles between the horizontal direction and the four curves formed by $Z_b$-direction velocity peaks were also changed from 90° to

---

---

**Data availability statement:** All relevant data are within the paper and its Supporting Information files.

**Funding:** The authors acknowledge the support provided by the National Natural Science Foundation of China (No. 52275257, No. 51705264) and the Innovative Training Program for College Students - 202410298024Z

**Competing interests:** The authors declare no conflict of interest.

72°, 69°, 61°, and 56° respectively at the flight speed of 3 m/s. Therefore, the installation of the nozzle and the formulation of the spray strategy are crucial to improve the spray deposition effect.

## 1. Introduction

Multi-rotor plant protection Unmanned Aerial Vehicle (UAV), which has broken through the constraint factors of planting terrains and crop types, has the ability that traditional ground self-propelled and manual plant protection equipment cannot complete [1–6]. Therefore, due to the vast territory and diverse planting terrain, multi-rotor plant protection UAVs have been widely used in China, especially four-rotor and hexa-rotor UAVs [7,8]. By the end of 2021, the number of multi-rotor plant protection UAVs is about 120000 in China, and the annual spray area is about 1070000000 mu. Up to now, multi-rotor plant protection UAV has become one of the indispensable plant protection equipment in China [9,10].

The research results show that the flight-coupled wind field composed of upwind airflow, downwash of the rotors, and natural crosswinds will significantly affect the spray deposition effect of multi-rotor plant protection UAV. The horizontal coverage width, vertical wind speed magnitude, and spatiotemporal evolution of the flight-coupled wind field are the critical factors that affect the deposition level and spray coverage of the target [7,11,12]. Deposition uniformity and drift under the flight-coupled wind field are typical problems faced by aviation spray (as shown in Fig 1). Under the downforce of the flight-coupled wind field, the droplet groups are transported to the crop canopy area, and the crop canopy moves laterally. If the deposition of droplet groups on the canopy and the lateral movement of crops occurred simultaneously, the uniformity of deposition on both sides of leaves would be significantly improved [13]. So the flight-coupled wind field is a key factor affecting the deposition and drift levels of the multi-rotor plant protection UAV [14]. Therefore, related studies have been conducted by scholars on the distribution characteristics of the flight-coupled wind field for multi-rotor plant protection UAVs, and the relationship between airflow distribution and droplet deposition. Of course, analyzing the dynamic distribution of the flight-coupled wind field during flight is the first crucial step.

Combining CFD numerical calculation with a wind speed test could well obtain the spatiotemporal characteristics of the flight-coupled wind field and analyze the causes of the flow law [4,10]. The evolution law of wake vortices for the Thrush 510G over time, and the interaction between wake vortices and hard ground were simulated by Bin Zhang et al. [15] using the CFD method. Based on the gas-liquid multiphase flow model, the CFD technique was applied to analyze the wake vortices and the droplet groups motion trajectory for the Thrush 510G aircraft by Bin Zhang et al. [16]. Although the authors' research is only based on the two-dimensional model, we can observe the distribution details of the wake vortices and the motion trajectories of the droplet groups in real time, which is impossible for the wind speed test. The aerodynamic interference characteristics between the tail-wing and the propeller-rotor, and

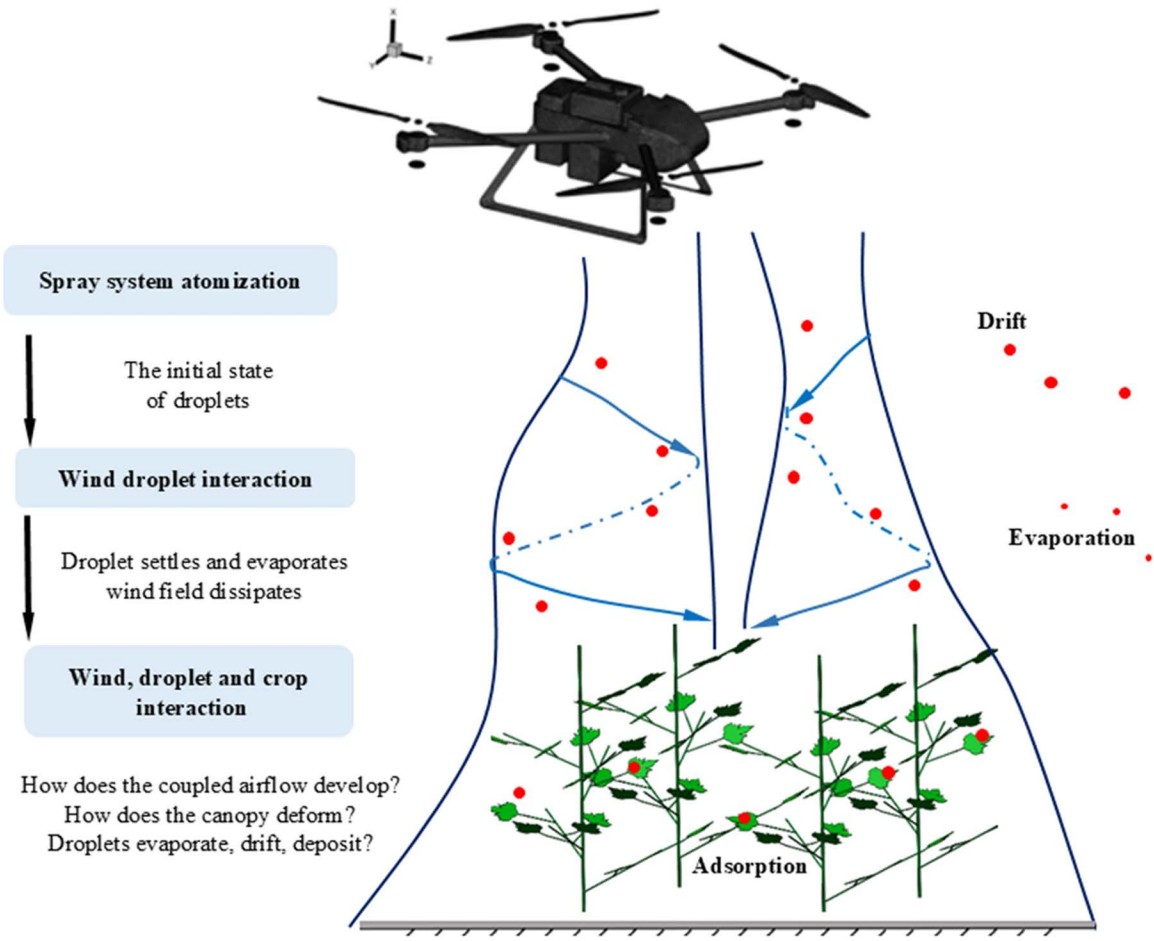

**Fig 1. Schematic diagram of flight spray of four-rotor plant protection UAV.** In terms of parameter measurement of downwash airflow for plant protection UAV, scholars have also conducted many effective studies. Setting the wireless micro-meteorology airflow speed sensors in a line, the flight-coupled wind field of a single-rotor plant protection UAV working in rice field was measured by Wang et al. [19]. According to the results of the tests, the conclusion could be drawn that the flight direction, flight height, and natural crosswind weakened the downwash intensity in the vertical direction. Wang et al. [20] measured the velocity of the flight-coupled wind field in $x$, $y$, and $z$ directions based on the application scenario of hybrid rice breeding supplementary pollination. The test experiment indicated that the relationship among the speed values was $V_x$ (flight direction) > $V_y$ (perpendicular to the $xoz$ plane) > $V_z$ (the vertical direction), and the airflow in the flight direction was the most useful to the supplementary pollination. Li et al. [21] set up three-direction airflow speed sensors in a line in the rice canopy, and then obtained the airflow velocity at several heights in the rice canopy. The results indicated that a "steep" effect existed in the velocity values at the UAV flight direction, which meant that the increasing rate of the speed value in the flight direction was higher than the reducing rate in the backward direction.

its effects on the lateral stability were researched for a tilt-rotor UAV by Jung et al. [17] during transient flight modes. The downwash of a hexa-rotor plant protection UAV in hover was calculated by Zhang et al. [18] in a virtual orchard, and the distribution laws of the downwash were analyzed in different natural crosswind speeds, fruit tree growth stages, and hover heights. The interaction mechanism between the flight-coupled wind field of a UAV and the vortex in the rice canopy was researched by Li et al. [11]. In the research results, to enhance the target precision of spray for the UAV, the theoretical model including the UAV motion parameters and the vortex movement was also established [11].

The flight-coupled wind field forces the droplet groups to move, deposit, and drift; the flight-coupled wind field drives the crop canopy to bend; then the strong flight-coupled wind field develops into a faint wind. In conclusion, the flight-coupled

wind field is a crucial factor for the multi-rotor plant protection UAV spray. However, research on the flight-coupled wind field of the multi-rotor plant protection UAV mainly focuses on the spatial scale, and the detailed flow laws of flight-coupled wind field in the temporal scale are rarely mentioned. In addition, many studies on downwash and flight-coupled wind field based on the CFD tools are limited to the rotor structure itself. The scientific research achievements of downwash and flight-coupled wind field based on the overall multi-rotor plant protection UAV are rarely mentioned. In the field of plant protection, the CFD tool was initially used for the auxiliary analysis of orchard spray [22,23]. However, the CFD tool has continued to be used in aviation spray in the past decade. Therefore, this paper took the overall structure of a four-rotor plant protection UAV as the study object, and studied the evolutionary law of flight-coupled wind field in the spatiotemporal dimensions by combining Reynolds Averaged Navier-Stokes equations, Re-Normalization Group $k$-$\varepsilon$ turbulence model, dynamic mesh based on the spring smoothing and layering method, pressure-velocity coupling algorithm. The $Z_b$-direction wind speed test experiment was designed to measure the airflow velocities of the flight-coupled wind field at the observation points for the four-rotor plant protection UAV, and the reliability of the CFD simulation was confirmed by fitting the simulated and measured values of $Z_b$-direction wind speed. The workflow of this paper is shown in Fig 2.

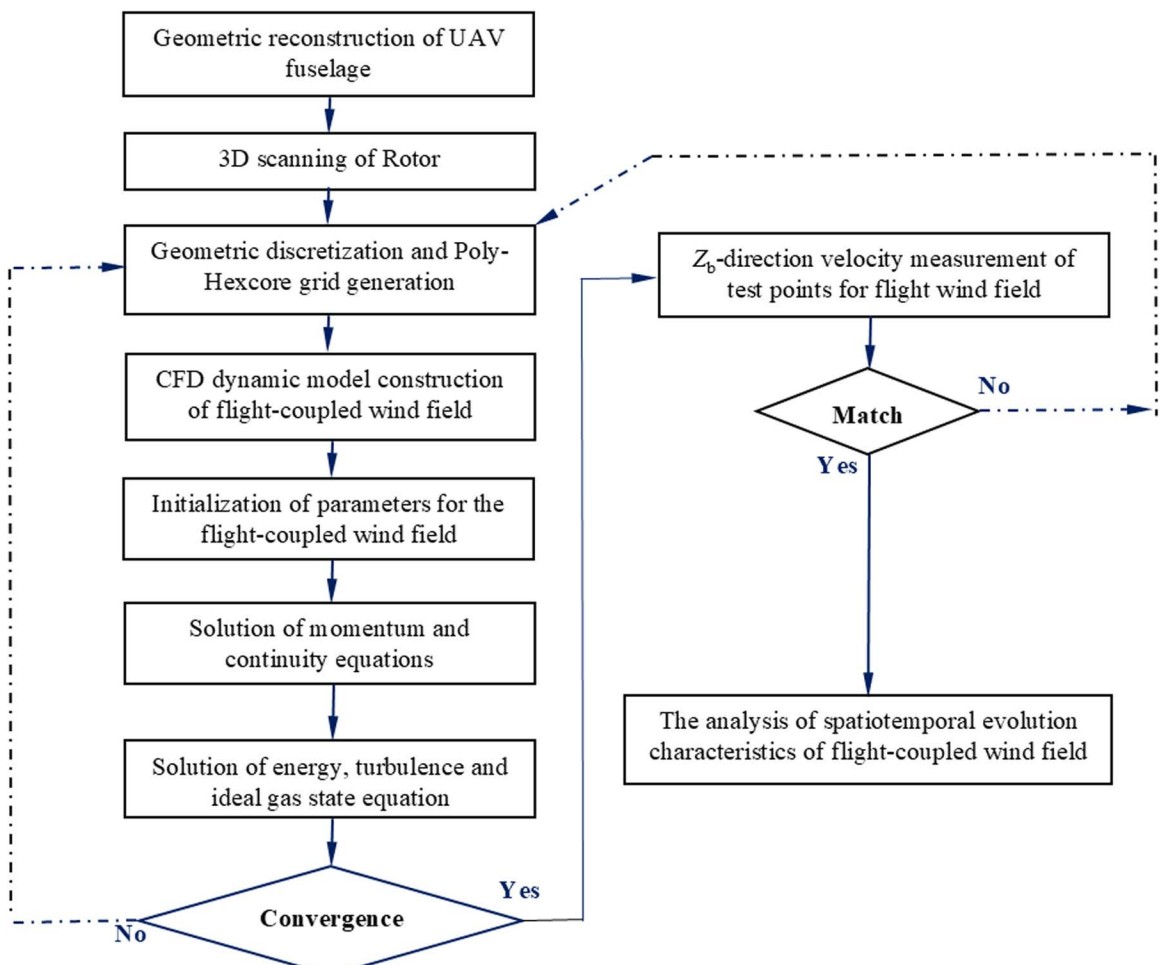

**Fig 2. Roadmap of this paper.**

## 2. CFD Models

### 2.1 Governing equations

In this paper, the governing equation of the flight-coupled wind Field was the Reynolds-Averaged Navier-Stokes (RANS) for the four-rotor plant protection UAV. Due to the rotation of UAV rotors, a source term was added to the governing equation, which could be written as follows [24,25]:

$$\frac{\partial}{\partial t}\iiint_{\partial V}\vec{W}\,dV + \iint_{\partial S}\left(\vec{F}-\vec{G}\right)\cdot\vec{n}\,dS = \iiint_{\partial V}\vec{Q}\,dV$$

(1)

$$\vec{W}=\begin{bmatrix}\rho\\\rho u\\\rho v\\\rho\omega\\\rho E_r\end{bmatrix},\ \vec{F}=\begin{bmatrix}\rho(\vec{q}-\vec{q}_w)\\\rho u(\vec{q}-\vec{q}_w)+p\vec{i}_x\\\rho v(\vec{q}-\vec{q}_w)+p\vec{i}_y\\\rho\omega(\vec{q}-\vec{q}_w)+p\vec{i}_z\\\rho H(\vec{q}-\vec{q}_w)\end{bmatrix},\ \vec{G}=\begin{bmatrix}0\\\tau_{xx}\vec{i}_x+\tau_{yx}\vec{i}_y+\tau_{zx}\vec{i}_z\\\tau_{xy}\vec{i}_x+\tau_{yy}\vec{i}_y+\tau_{zy}\vec{i}_z\\\tau_{xz}\vec{i}_x+\tau_{yz}\vec{i}_y+\tau_{zz}\vec{i}_z\\\Phi_x\vec{i}_x+\Phi_y\vec{i}_y+\Phi_z\vec{i}_z\end{bmatrix},\ \vec{Q}=\begin{bmatrix}0\\-\rho\omega\Omega\\0\\\rho u\Omega\\0\end{bmatrix}$$

(2)

where $V$ is the control volume of discrete unit in the computing domain; $S$ is the control surface of discrete unit in the computing domain; $\vec{W}$ is the conserved variable of the RANS equation; the $\vec{F}$ is the convective flux of the RANS equation; $\vec{G}$ is the diffusion flux of the RANS equation; $\vec{Q}$ is the source flux due to the rotation of the rotors; $H$ and $E_r$ are the total enthalpy and total internal energy of a discrete unit respectively; $\tau_{xx/yy/zz}$, $\tau_{xy/xz/yz}$, and $\Phi_{x/y/z}$ are the viscous quantities and heat fluxes in three directions respectively, $\vec{q}_w$ denotes grid velocity perpendicular to the unit surface, $\vec{q}$ denotes the air velocity vector; $u$, $v$, and $w$ are the air velocity components of a discrete unit in three directions respectively; $\rho$ and $p$ are the airflow density and pressure of a mesh unit; $t$ is the current calculation time; $\Omega$ is the rotational speed of the rotor.

### 2.2 Viscous Model

The Re-Normalization Group $k$-$\varepsilon$ turbulence model [26] was adopted to calculate turbulent effects caused by large shear flow, and the turbulent kinetic energy $k$ and turbulent dissipation rate $\varepsilon$ are shown in Equations 3 and 4.

Turbulent kinetic energy $k$:

$$\frac{\partial\left(\rho k\right)}{\partial t}+\frac{\partial\left(\rho k u_i\right)}{\partial x_i}=\frac{\partial}{\partial x_j}\left[\alpha_k\mu_{eff}\frac{\partial k}{\partial x_j}\right]+G_k+G_b-\rho\varepsilon$$

(3)

Turbulent dissipation rate $\varepsilon$:

$$\frac{\partial\left(\rho\varepsilon\right)}{\partial t}+\frac{\partial\left(\rho\varepsilon u_i\right)}{\partial x_i}=\frac{\partial}{\partial x_j}\left[\alpha_\varepsilon\mu_{eff}\frac{\partial\varepsilon}{\partial x_j}\right]+C^*_{1\varepsilon}\frac{\varepsilon}{k}G_k-C_{2\varepsilon}\rho\frac{\varepsilon^2}{k}$$

(4)

$$C^*_{1\varepsilon}=C_{1\varepsilon}-\frac{\eta\left(1-\eta/\eta_0\right)}{1+\beta\eta^3}$$

(5)

$$\eta=\sqrt{2E_{ij}\cdot E_{ij}}\frac{k}{\varepsilon}$$

(6)

$$E_{ij} = \frac{1}{2} \left( \frac{\partial u_i}{\partial x_j} + \frac{\partial u_j}{\partial x_i} \right)$$

(7)

where $\rho$ is the airflow density of a mesh unit; $k$ and $\varepsilon$ are the turbulent kinetic energy and the turbulent dissipation rate; $t$ is the current calculation time; $u_i$ is the time mean velocity in the direction $i$; $x_i$ and $x_j$ are coordinates in the $i$ and $j$ directions; $\alpha_k$ and $\alpha_\varepsilon$ are turbulence model constants, $\alpha_k = \alpha_\varepsilon = 1.39$; $\mu_{eff}$ is total viscosity of airflow, $\mu$ is kinematic viscosity of airflow, $\mu_t$ is turbulent viscosity of airflow, $\mu_{eff} = \mu + \mu_t$, $\mu_t = \rho C_\mu \frac{k^2}{\varepsilon}$; $G_k$ is the production term of turbulent kinetic energy $k$ due to the average velocity gradient; $G_b$ is the production term of turbulent kinetic energy $k$ caused by buoyancy; $C_{1\varepsilon}^*$ is an incremental modification of the RNG $k$-$\varepsilon$ equation based on the turbulent kinetic energy $k$ and turbulent dissipation rate $\varepsilon$; $\eta$ is the ratio of turbulence to mean flow in the time scale; $E_{ij}$ is the characteristic strain rate; $C_\mu$, $C_{1\varepsilon}$, $\eta_0$, $\beta$, $C_{2\varepsilon}$ are constant terms, $C_\mu = 0.0845$, $C_{1\varepsilon} = 1.42$, $\eta_0 = 4.377$, $\beta = 0.012$, $C_{2\varepsilon} = 1.68$.

### 2.3 Structure, Boundary Condition, and Mesh Model

**2.3.1 Structure.** In this research, the flight-coupled wind field testing experiment has been strongly supported by XAIRCRAFT (Guangzhou, China). The 3D model construction process of the four-rotor plant protection UAV is shown in the following Fig 3. A Solidworks drawing tool was used to draw the main structure (as shown in Fig 3b). The point cloud of the execution component rotor was obtained by a 3D Optimscan 5–2015011 K05 scanner (as shown in Fig 3c). The Solidworks tool was used to synthesize the cloud point, and the complex surface of the UAV rotor was obtained (as shown in Fig 3e). Since the rotors are closer to the main structure, the flight-coupled wind field would be significantly affected by the UAV body structure, the main structure and rotors should be assembled for the transient calculation of the flight-coupled wind field (as shown in Fig 3f).

**2.3.2 Boundary Condition Setting.**

**(1) *Motion Conditions for the UAV:*** The motion system diagram of the four-rotor plant protection UAV is shown in the following Fig 4. It can be seen from Fig 4 that to maintain the torque balance of the UAV body, the rotational speed between adjacent rotors is opposite. Rotor 1 and Rotor 3 are set to rotate clockwise, while Rotor 2 and Rotor 4 are set to rotate counterclockwise. When the rotational speeds of the four rotors are equal, and the resultant force generated by the rotation of the four rotors is exactly equal to the weight of the UAV body, the UAV will be in the hover state. The adjustment of flight speed and flight attitude can be made by changing the rotation speed of the four rotors. There were two coordinate systems in Fig 4, namely the absolute coordinate system $O_bX_bY_bZ_b$ and the relative coordinate system $O_eX_eY_eZ_e$; and the coordinate systems were set according to the right-hand rule. The geometric center of each rotor was the coordinate origin of the respective relative coordinate system, and the three coordinate axes were parallel to the coordinate axes of the same name in the absolute coordinate system. The absolute coordinate system was used to set the position and rotation angular speed of each rotor, and the relative coordinate system was used to set the rotation axis of each rotor. When the four-rotor plant protection UAV flew along the negative direction of the $X_b$-axis in the absolute coordinate system, the flight speed was 3m/s. When flying, the overall forward inclination of the UAV body was 9°, so the rotation axis of each rotor was not the $Z_e$-axis of the relative coordinate system. The rotation axis of each rotor was in the $O_eX_eZ_e$ plane of the relative coordinate system, and the coordinate values of the $X_e$-axis and $Z_e$-axis were -0.156434 and 0.987688.

In this paper, to better match the wind field test and find the evolution law of the dynamic flight-coupled wind field, the dynamic theoretical calculations of the flight-coupled wind field under two working conditions were carried out. The first working condition was that the four-rotor plant protection UAV hovered for 1.3 s, and then flew in the $X_b$-axis negative direction at a speed of 3 m/s. The second working condition was that the UAV flew directly along the $X_b$-axis negative direction at a speed of 3 m/s. In addition, in the two working conditions, the working parameters of each rotor are shown in Tables 1 and 2 below.

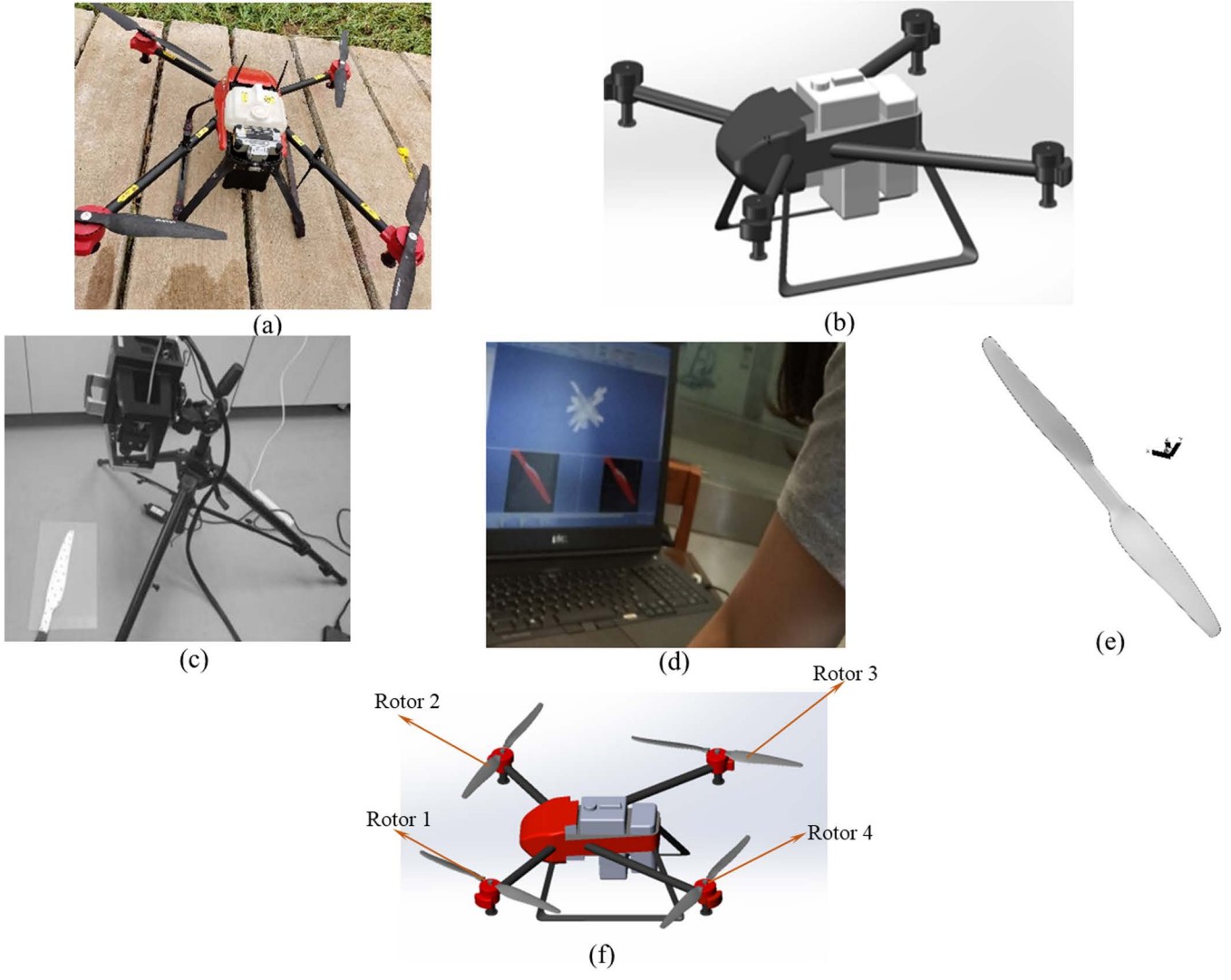

**Fig 3. Scanning and reconstruction of the four-rotor UAV. (a)** The actual structure of the four-rotor UAV. **(b)** Geometric reconstruction of the four-rotor UAV fuselage. **(c)** Calibrating and scanning the rotor. **(d)** Point cloud synthesis site. **(e)** View of the structured model of the rotor component. **(f)** The final 3D model of the four-rotor UAV.

It was worth noting that due to the dynamic calculation, the working parameters of each rotor would be significantly different when the four-rotor plant protection UAV was fixed and flying. When the time was 0 s, the rotation center and rotational angular velocity of the same rotor were the same in the Working Conditions 1 and 2. In Working Condition 1, when the UAV body was fixed, the rotation center of each rotor remained unchanged; when flying at a speed of 3 m/s along the negative direction of the $X_b$-axis, the rotation center $X_b$-coordinate of each rotor continuously decreased, and the relationship between the $X_b$-coordinate and the calculation time $t$ was shown in Table 1. In Working Condition 2, the $X_b$-coordinate of the rotation center of each rotor kept decreasing, as shown in Table 2.

**(2) *The connection method for calculating regions:*** As can be seen from Fig 5, the entire computing area was divided into four computing domains. The Computing Domain 1 was a region of four small cylinders, each enclosing a rotor; the Computing Domain 2 was the hexahedral region that wrapped the UAV body; Computing Domain 3 was the hexahedron region above that moved with the UAV; Computing Domain 4 was the stationary hexahedron region close to the ground.

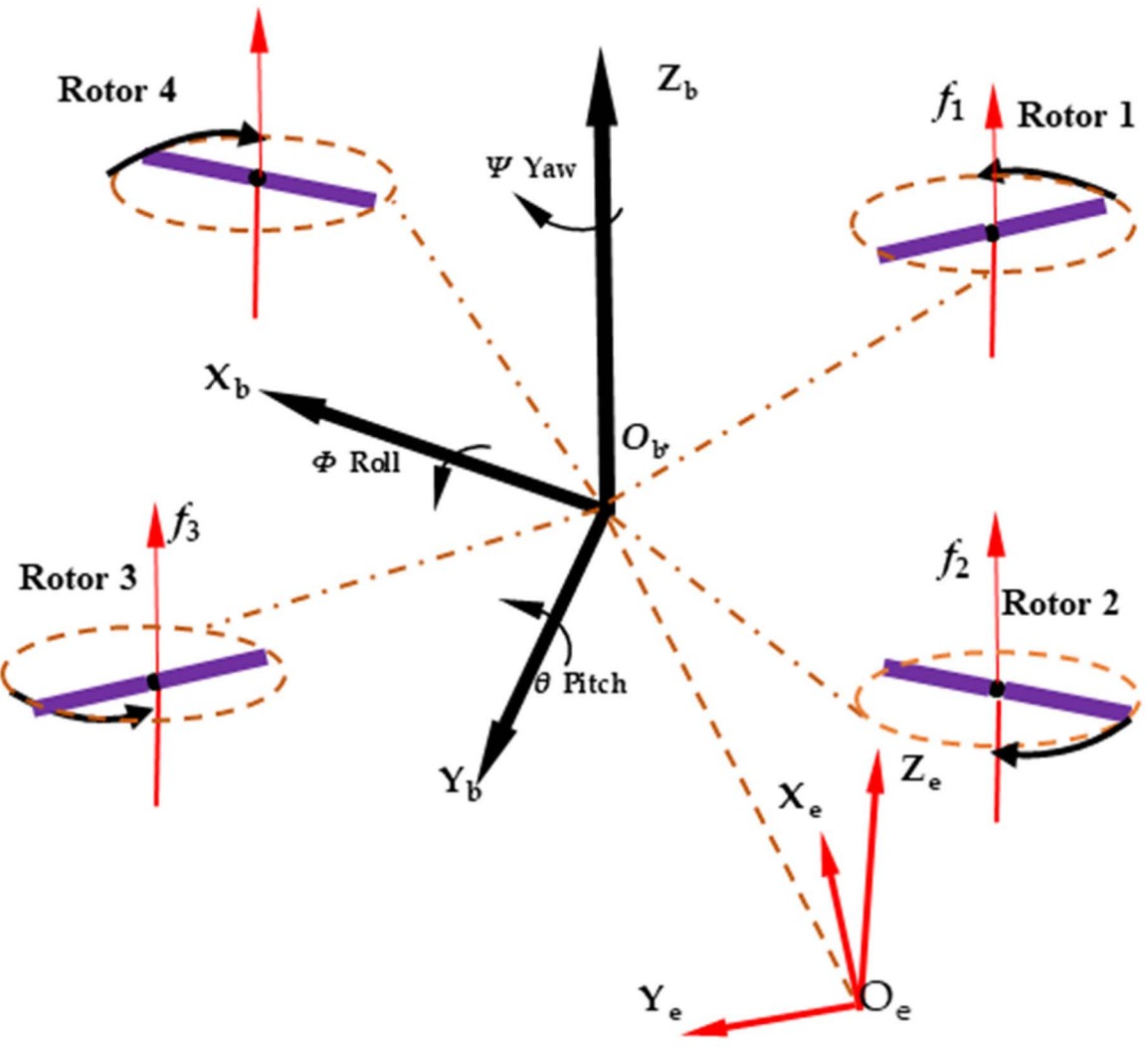

**Fig 4. Four-rotor UAV motion system diagram.**

**Table 1. Running parameters of each rotor in Working Condition 1.**

| Rotor number | Coordinates of the rotation center in Working Condition 1 | | | | Angular velocity (rad/s) |
|---|---|---|---|---|---|
| | x/ m | | y/ m | z/ m | |
| | | $t \le 1.3$(s) (UAV Fixed) | $t > 1.3$(s) (UAV Flight) | | |
| 1 | -0.415239839 | -0.415239839-(3t-3.9) | -0.518860346 | -0.099685111 | 249.9 |
| 2 | -0.415239839 | -0.415239839-(3t-3.9) | 0.518860347 | -0.099685111 | -249.9 |
| 3 | 0.541537512 | 0.541537512-(3t- 3.9) | 0.516302113 | 0.051853534 | 253 |
| 4 | 0.541537512 | 0.541537512-(3t- 3.9) | -0.516302112 | 0.051853534 | -253 |

Boolean operation was performed on the cylinder and rotor in Computing Domain 1, and the solid domain of the rotor was deleted. In Computing Domain 2, a Boolean operation was performed on the hexahedron and the UAV body, and the solid domain of the UAV body was deleted; an area that exactly corresponds to the spatial location of the Computing

**Table 2. Running parameters of each rotor in Working Condition 2.**

| Rotor number | Coordinates of the rotation center in Working Condition 2 | | | Angular velocity (rad/s) |
|---|---|---|---|---|
| | *x*/ m (UAV Flight) | *y*/ m | *z*/ m | |
| 1 | -0.415239839-3*t* | -0.518860346 | -0.099685111 | 249.9 |
| 2 | -0.415239839-3*t* | 0.518860347 | -0.099685111 | -249.9 |
| 3 | 0.541537512-3*t* | 0.516302113 | 0.051853534 | 253 |
| 4 | 0.541537512-3*t* | -0.516302112 | 0.051853534 | -253 |

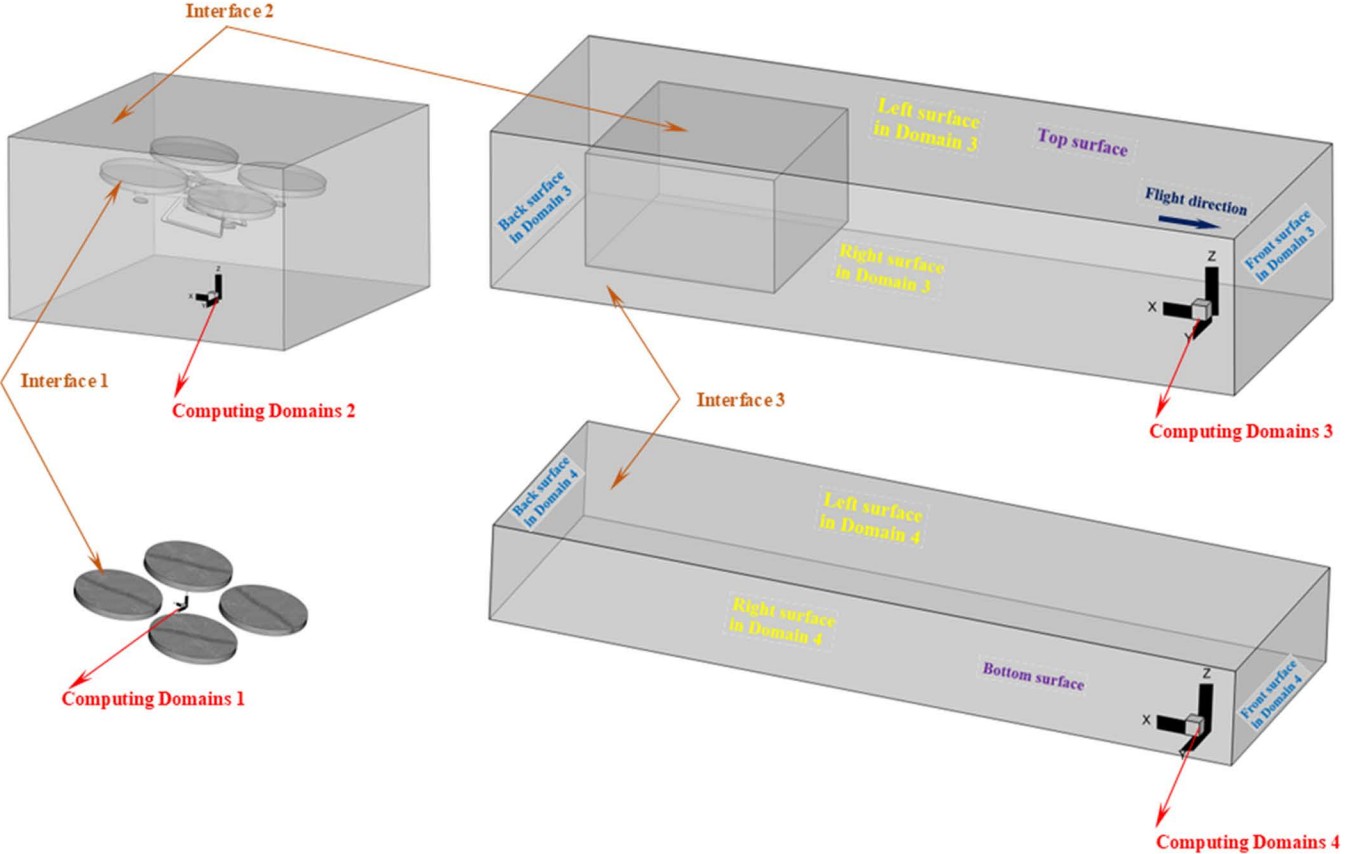

**Fig 5. Calculating regions connections and boundary surfaces markings.**

Domain 1 was deleted in Computing Domain 2, then Computing Domain 1 and Computing Domain 2 were connected and airflow data were exchanged through Interface 1. An area that exactly corresponds to the spatial location of Computing Domain 2 was deleted in the third computational domain, then Computing Domain 2 and Computing Domain 3 were connected and airflow data were exchanged through Interface 2. The lower-end face in Computing Domain 3 was completely aligned with the upper-end face in Computing Domain 4, then Computing Domain 3 and Computing Domain 4 were connected and airflow data were exchanged through Interface 3. In dynamic calculation, to ensure that the lower-end face of the third computing domain and the upper-end face of the fourth computing domain were always aligned, the surface directly opposite the negative direction of the $X_b$-axis in Computing Domain 3 was the grid vanishing surface, and the

surface directly opposite the positive direction of the $X_b$-axis was the grid generating surface, and both of the grid generation and disappearance rates were 3 m/s.

In general, through the setting of the above connection method, the dynamic numerical calculation of the flight-coupled wind field for the four-rotor plant protection UAV could be realized.

**(3) Boundary Conditions for CFD Models:** The background operating pressure of the flight-coupled wind field CFD model was 101325 Pa. The initial condition ($t=0$ s) of the overall CFD model was that gauge pressure $p=0$ Pa, temperature $T=300$ K, three velocity components $u=v=w=0$. The boundary conditions ($t\geq0$ s) for the outermost surfaces of the CFD model (at Computing Domains 3 and 4) and the solid surfaces inside the CFD model (at Computing Domains 1 and 2) in Fig 5 were set as shown in the following Table 3.

**2.3.3 Mesh Model.** The surface grid distribution law for the four-rotor plant protection UAV is shown in Fig 6. To show the rotor surface grid more clearly, the grid distribution near the area was enlarged.

The meshes for all the computing domains were constructed in ANSYS Fluent Meshing (ANSYS ANSYS 2023R1, Inc., USA), and the mesh models of Computing Domains 1, 2, 3 and 4 were constructed using the Poly-Hexcore [27] mesh strategy. To improve the calculation accuracy of the flight-coupled wind field, the minimum and maximum side length sizes were finally set at 0.003 m and 0.05m respectively, the growth rate of grid side length was set at 1.05, the curvature normal angle of the complex surface was set at 10 degrees, the number of grid cells at small gaps was set to 3, and the grid number of the entire computing domain eventually reached 4.598 million. The results showed that the maximum orthogonal quality parameters of surface mesh and volume mesh are 0.49 and 0.63 (much less than 1) respectively, which met the grid quality requirements.

## 2.4 Numerical Solution Method

The finite volume method was adopted to discretize the RANS equation, and The CFD simulations were carried out by the ANSYS Fluent Double-Precision Solver(ANSYS 2023R1, Inc., USA) using RYZEN with AMD 9554 (64 cores, two pieces; 3.75 GHz; 768 GB RAM). A pressure-based solver was used to solve the flight-coupled wind field of the four-rotor plant protection UAV, and the coupled algorithm [26] was used to calculate pressure-velocity coupling for the unsteady simulations. The pressure was discretised with a second-order scheme. The density, momentum and energy were discretised with a second-order upwind scheme. The turbulence quantities were discretised with a first-order upwind scheme. The transient term was discretised with a second-order implicit scheme. The convergence criteria was that the residuals of all variables were set as $1.0\times10^{-4}$.

**Table 3. Setting of boundary conditions.**

| Boundary surface name | Boundary type in ANSYS Fluent | Boundary type expressed in mathematical expressions |
|---|---|---|
| Front surface in Domain 3 | Pressure Outlet | $p=0$ |
| Front surface in Domain 4 | | |
| Back surface in Domain 3 | | |
| Back surface in Domain 4 | | |
| Left surface in Domain 3 | | |
| Left surface in Domain 4 | | |
| Right surface in Domain 3 | | |
| Right surface in Domain 4 | | |
| Top surface | | |
| Bottom surface (ground) | Wall (No Slip) | $u=v=\omega=0,$ $\frac{\partial p}{\partial x}=\frac{\partial p}{\partial y}=\frac{\partial p}{\partial z}=0$ |
| UAV fuselage surfaces | | |
| UAV rotor surfaces | | |

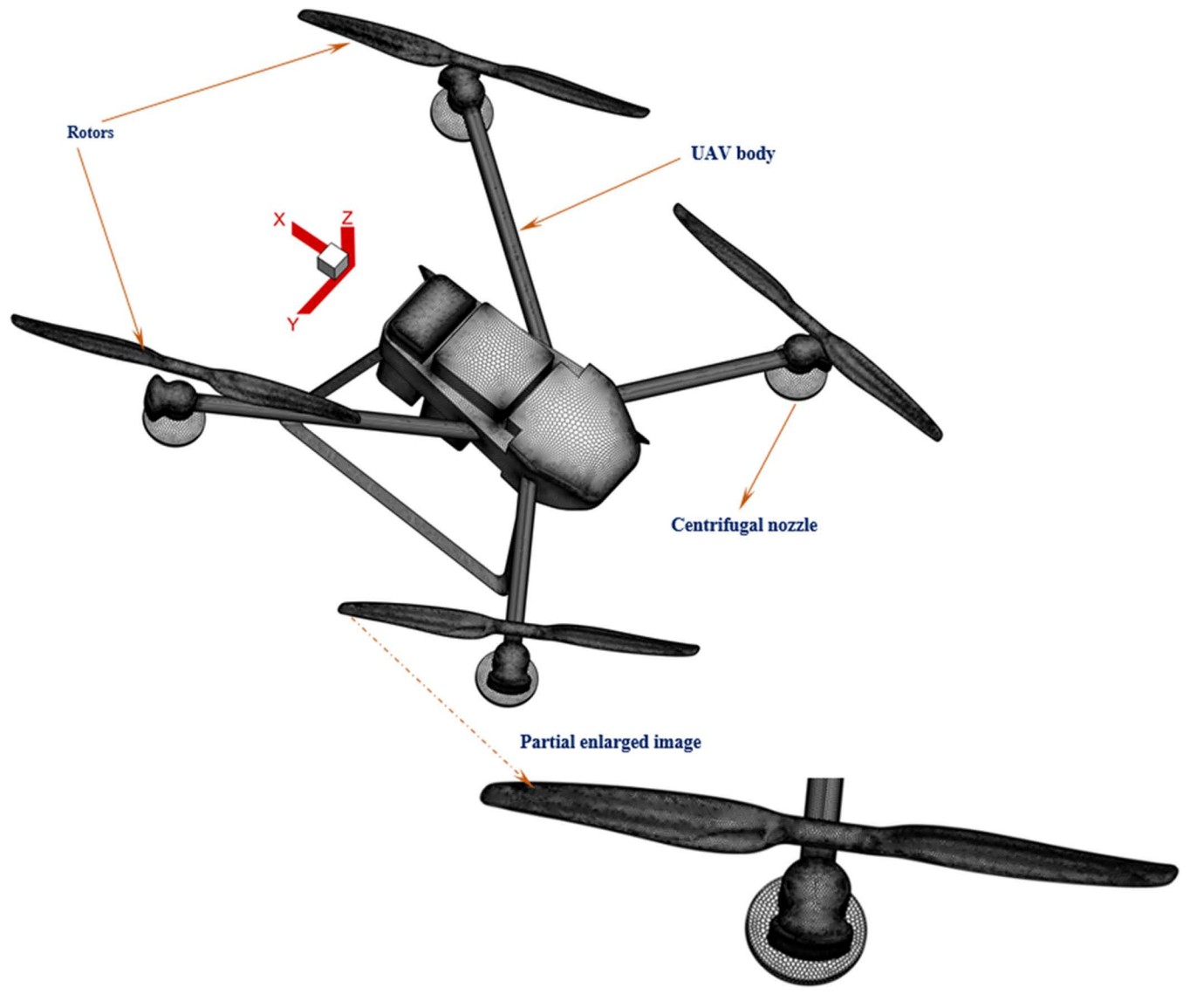

**Fig 6. Grid structure diagram for the solid surfaces of the four-rotor UAV.**

## 3. The Evolution and Distribution Law of the Flight-Coupled Wind Field

### 3.1 Feasibility Verification of the Flight-Coupled Wind Field Theoretical Simulation

To obtain a reliable dynamic distribution law of flight-coupled wind field, feasibility verification of theoretical calculation is a prerequisite before detailed analysis. In this paper, the flight-coupled wind field speed measurement and calculation were strongly supported by XAIRCRAFT (Guangzhou, China). The flight-coupled wind field testing site is shown in Fig 7a. The Bluetooth wireless wind velocity sensor (UNI-T) was applied to record the $Z_b$-axis wind speed values of testing points (as shown in Figs 7a and 7d), and the sampling frequency of the UNI-T sensor was 1 Hz. The test points were set right under the rotor at the four-rotor plant protection UAV flight path, the height of the test points from the rotor was 2.25 m, as shown in Fig 7d and 7e. The flight direction of the four-rotor plant protection UAV was defined as the $X_b$-axis negative direction,

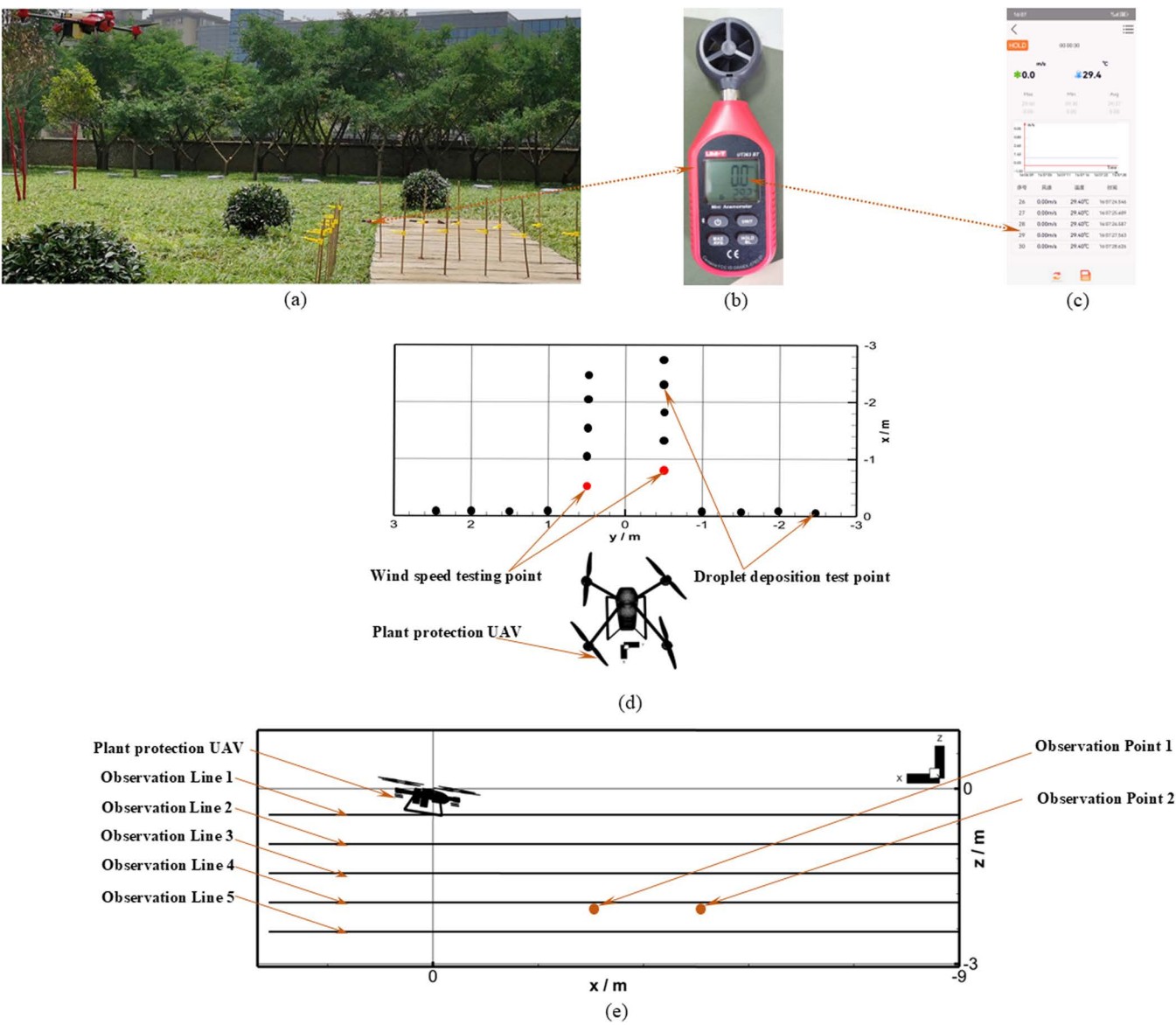

**Fig 7. Wind field test and calculation observation position setting. (a)** Flight-coupled wind field testing site. **(b)** Bluetooth wireless anemometer. **(c)** Bluetooth data receiving terminal. **(d)** Layout of testing points of flight-coupled wind field (spray). **(e)** Layout diagram of observation lines (points) for dynamic calculation of flight-coupled wind field.

the direction straight down was defined as the $Z_b$-axis negative direction, and the relationship between the $X_b$-axis, $Y_b$-axis, and $Z_b$-axis satisfied the right-hand rule, as shown in Fig 7e. The observation points (lines) for the dynamic calculation of flight-coupled wind field were also illustrated in Fig 7e. According to the data provided by XAIRCRAFT, the flight speed of the four-rotor plant protection UAV was defined as 3 m/s, the forward tilt angle of the UAV body was set as 9°, the rotational angular velocity of the front two rotors was set as 249.9 rad/s, the rotational angular velocity of the rear two rotors was set as 253 rad/s. It took some time for the spatial distribution of the flight-coupled wind field to become stable after take-off, so the numerical calculations of the flight-coupled wind field under two working conditions were carried out. Based on Working Condition 1, the flow evolution law during the transition from hover state to flight state was mainly

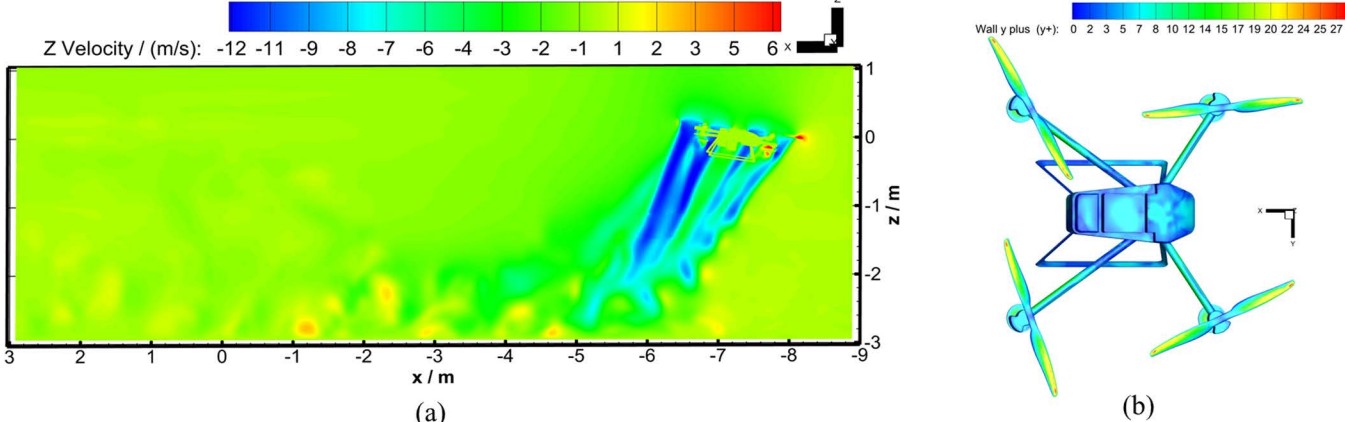

**Fig 8. Numerical calculation results in flight wind field. (a)** $Z_b$-axis velocity distribution map at 2.492 s of the flight-coupled wind field for Working Condition 2. **(b) y+** sketch map for Working Condition 2 at 2.492 **s.**

studied; the flight-coupled wind field distribution law after flight stabilization was analyzed in particular in Working Condition 2.

The calculated value of $y+$ is an important indicator of whether the thickness of the mesh boundary layer (the first layer of mesh from the solid wall to the fluid) is reasonable. When the calculated value of $y+$ is reasonable, the development and diffusion law of the flow field can be confirmed to be reasonable. The expression for $y+$ is as follows [28]:

$$y+ = \frac{\Delta y}{\mu} \cdot \sqrt{\frac{\tau_w}{\rho}}$$

(8)

where $y+$ is a dimensionless number that characterizes whether the first-layer grid is reasonable; $\Delta y$ is the thickness of the first boundary layer; $\rho$ and $\mu$ are the density and viscosity of the fluid within the first boundary layer mesh; $\tau_w$ is the shear stress on the solid wall of the first layer.

After the numerical calculations, the $Z_b$-axis velocity distribution for the longitudinal section passing through the rotor center and the y+ cloud map of the four-rotor plant protection UAV wall in the Working Conditions 2 were shown in Fig 8. The $Z_b$-axis velocities of wind speed Observation Point 1 in two working conditions, the $Z_b$-axis velocities of wind speed Observation Point 2 in two working conditions, and the $Z_b$-axis average velocity of two test points were all presented in Fig 9.

In Working Condition 1, the four-rotor plant protection UAV hovered for 1.3s, and the flight altitude was relatively low, so the ground effect had a great influence on the $Z_b$-axis wind speed value of Observation Points 1 and 2 (As shown in Section 3.2). In addition, the sampling frequency of the UNI-T wind speed sensor is relatively low (1 Hz). Therefore, only the $Z_b$-axis velocity peaks of the testing point in the wind field test and the observation point in Working Condition 2 could be used for comparison. When the flight-coupled wind field swept over the Observation Point 1, it was not completely stable. So the relative error of the wind speed peak in the $Z_b$-axis for the testing point and Observation Point 2 (in Working Condition 2) was 12.7%. So overall, the division of the grid was reasonable.

### 3.2 Evolution Law of the Flight-Coupled Wind Field in Spatial and Temporal Dimensions

Based on the above theoretical calculation method and flight conditions, the unsteady numerical calculations of Working Conditions 1 and 2 were carried out. The velocity distribution diagrams of the hovering wind field and flight-coupled wind field for Working Condition 1 are shown in Fig 10.

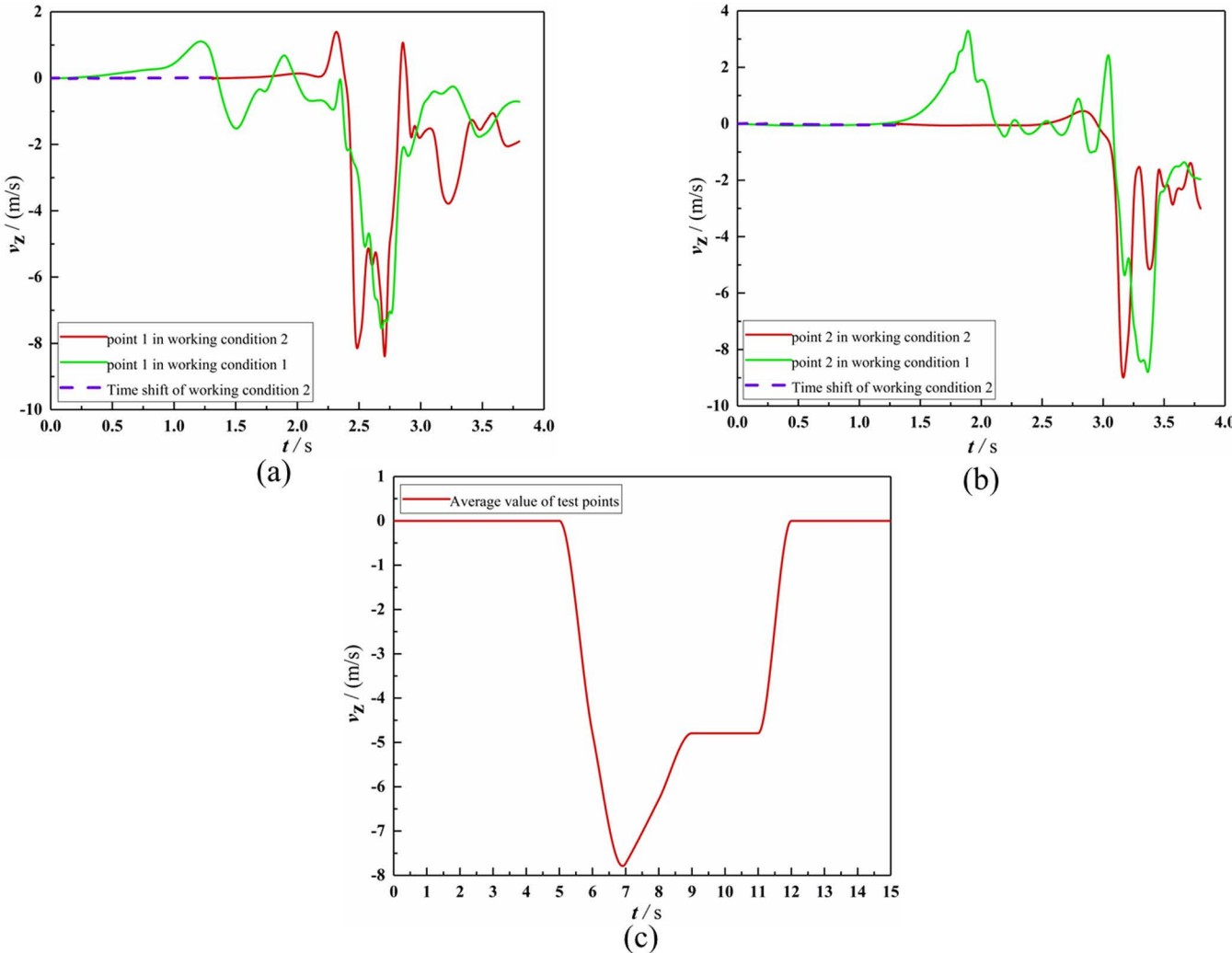

**Fig 9. Comparison between numerical calculation and experiment of $Z_b$-axis velocity at characteristic points. (a)** At Observation Point 1 in numerical calculations. **(b)** At Observation Point in numerical calculations. **(c)** The average value of two test points in the wind speed measurement experiment. From Fig 8b, it could be seen that after the flight was in dynamic stability, the calculated value of $y+$ on the surfaces of the four-rotor plant protection UAV was below 15. Only the calculated value of $y+$ in the large shear area on the upper-end surfaces of the rotors would reach 15 or more. During the flight, the calculated value of $y+$ in the vortex separation zone from the wing tip to one-third of the wing center would approach 30. In addition, to better illustrate the relative error between the experimental test (Testing Point) and theoretical calculation (Observation Point), Table 4 was drawn out.

In the velocity distribution diagrams, the vertical distance of the cross-section from the rotor is 0.9 m, the longitudinal section passes through the center of the rotor, and the longitudinal section is parallel to the flight direction. It could be seen that the speed of the four-rotor plant protection UAV fuselage surface unit was 0 m/s when the UAV was hovering, as shown in Fig 10a. When the UAV flew forward, the speed of the UAV fuselage surface was 3 m/s, which made the hovering downwash in the dynamic stable state suddenly interrupted laterally, as shown in Fig 10b. It could be also seen by combining Figs 10a and 10b that the lateral displacement between the four-rotor plant protection UAV body and the airflow field occurred, and the downwash airflow was thrown behind the UAV body. As could be seen from Figs 10a-10d, the evolution of flight-coupled wind field for the four-rotor plant protection UAV had undergone a dynamic process. During the process from hover to flight, the "downwash airflow" experienced lateral interruption, developed back

**Table 4. Comparison of peak vertical velocity values between test points and observation points.**

| Object | | Coordinates | | Peak vertical velocity | | Coordinates | | | Peak vertical velocity | Absolute value of peak velocity relative error |
|---|---|---|---|---|---|---|---|---|---|---|
| | | y/ m | z/ m | $V_z$-Peak/ (m/s) | Average value/ (m/s) | x/ m | y/ m | z/ m | $V_z$-Peak/ (m/s) | Relative error/ % |
| Wind field test | Testing Point 1 | -0.516302 | -2.25 | -8.0 | -7.9 | \ | \ | \ | \ | \ |
| | Testing Point 2 | 0.516302 | -2.25 | -7.7 | | \ | \ | \ | \ | \ |
| Calculation Working Condition 1 | Observation Point 1 | \ | \ | \ | \ | -2.536297 | -0.516302 | -2.25 | -7.5 | 5.1 |
| | Observation Point 2 | \ | \ | \ | \ | -4.536297 | -0.516302 | -2.25 | -8.7 | 10.1 |
| Wind field test | Testing Point 1 | -0.516302 | -2.25 | -8.0 | -7.9 | \ | \ | \ | \ | \ |
| | Testing Point 2 | 0.516302 | -2.25 | -7.7 | | \ | \ | \ | \ | \ |
| Calculation Working Condition 2 | Observation Point 1 | \ | \ | \ | \ | -2.536297 | -0.516302 | -2.25 | -8.4 | 6.3 |
| | Observation Point 2 | \ | \ | \ | \ | -4.536297 | -0.516302 | -2.25 | -8.9 | 12.7 |

into a dynamically stable state, and eventually became the "flight-coupled wind field" after flying for 0.696 s. During the process from hover to flight, the "flight-coupled wind field" distribution area of the cross-section developed from directly below the UAV body to gradually behind the UAV body, and finally maintainde a stable transverse distance from the UAV body. During the process from hover to flight, the "flight-coupled wind field" was influenced by the incoming airflow from the front, the four circular wind fields on the cross-section developed into four horseshoe tails, which rotated, and spread symmetrically backward across the cross-section in the $X_b$-axis negative direction. During the process from hover to flight, the "flight-coupled wind field" in the longitudinal section developed from relatively stable to the phenomenon that vortices constantly fell off in the opposite direction of flight. It was worth mentioning that the ground effect airflow spread along the direction of flight, and the $Z_b$-axis velocity value at the observation point was affected to some extent by this factor. This was why Working Condition 2, rather than Working Condition 1, was used to compare with the test when verifying the reliability of the numerical calculation.

The $Z_b$-axis velocity distribution comparison diagrams between the hover and flight phases of Working Condition 1 are given in Fig 11. By comparing Figs 11a-11f, it could be seen that four $Z_b$-axis velocity peaks appeared at Observation Lines 1–4 in the hovering and flight phase; the $Z_b$-axis velocity values of the two rotors ahead of the flight direction were smaller than the corresponding values of the two behind rotors. For the flight phase, when the vertical distance from the rotor gradually increased, the $Z_b$-axis wind speed peak gradually moved in the $X_b$-axis positive direction. During the flight phase, the dense atmosphere significantly woke the $Z_b$-axis wind speed peak of the front two rotors, and the $Z_b$-axis wind speed peak of the rear two rotors only decreased slightly. Based on Figs 10b and 11c, it could be seen that there was a significant lateral interruption in the downwash airflow after 0.245 s of flight, and the $Z_b$-axis wind speed peak significantly decreased due to the interruption.

Fig 12 shows the velocity streamline diagram of the transverse section (parallel to the horizontal direction) for the four-rotor plant protection UAV during flight. As can be seen from Fig 4, the front two rotors rotate outwards. Therefore, in forward flight, the air in front of the UAV body was compressed and flowed forward in general. Under the guidance of the front two rotors, the air near the front of the UAV body eventually moved forward and flowed to the rear side of the UAV

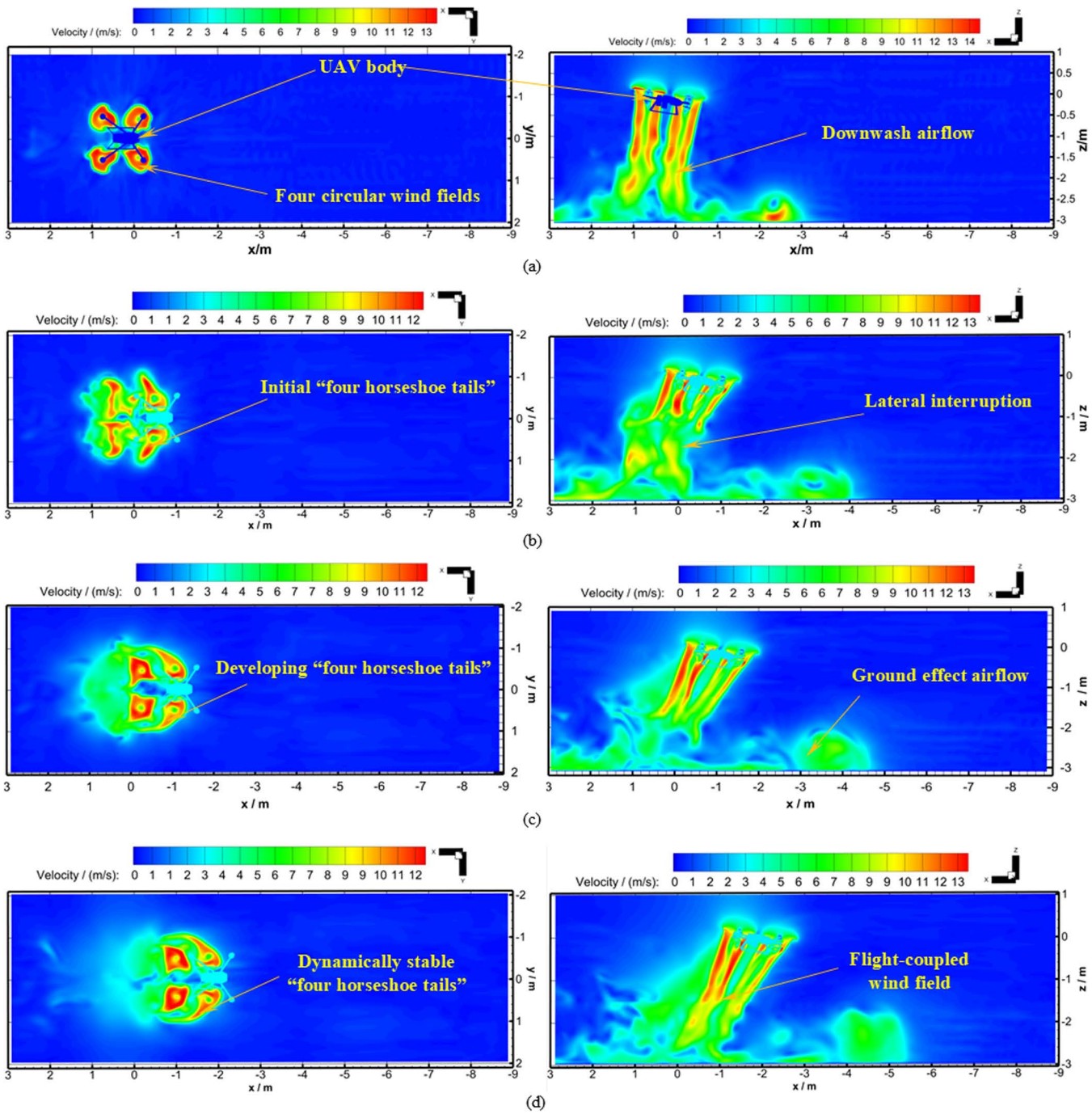

**Fig 10. Horizontal and vertical section velocity distribution diagram at typical flight moments for the UAV. (a)** After 1.3 s of hover (Working Condition 1). **(b)** After 0.245 s of flight (Working Condition 1). **(c)** After 0.446 s of flight (Working Condition 1). **(d)** After 0.696 s of flight (Working Condition 1).

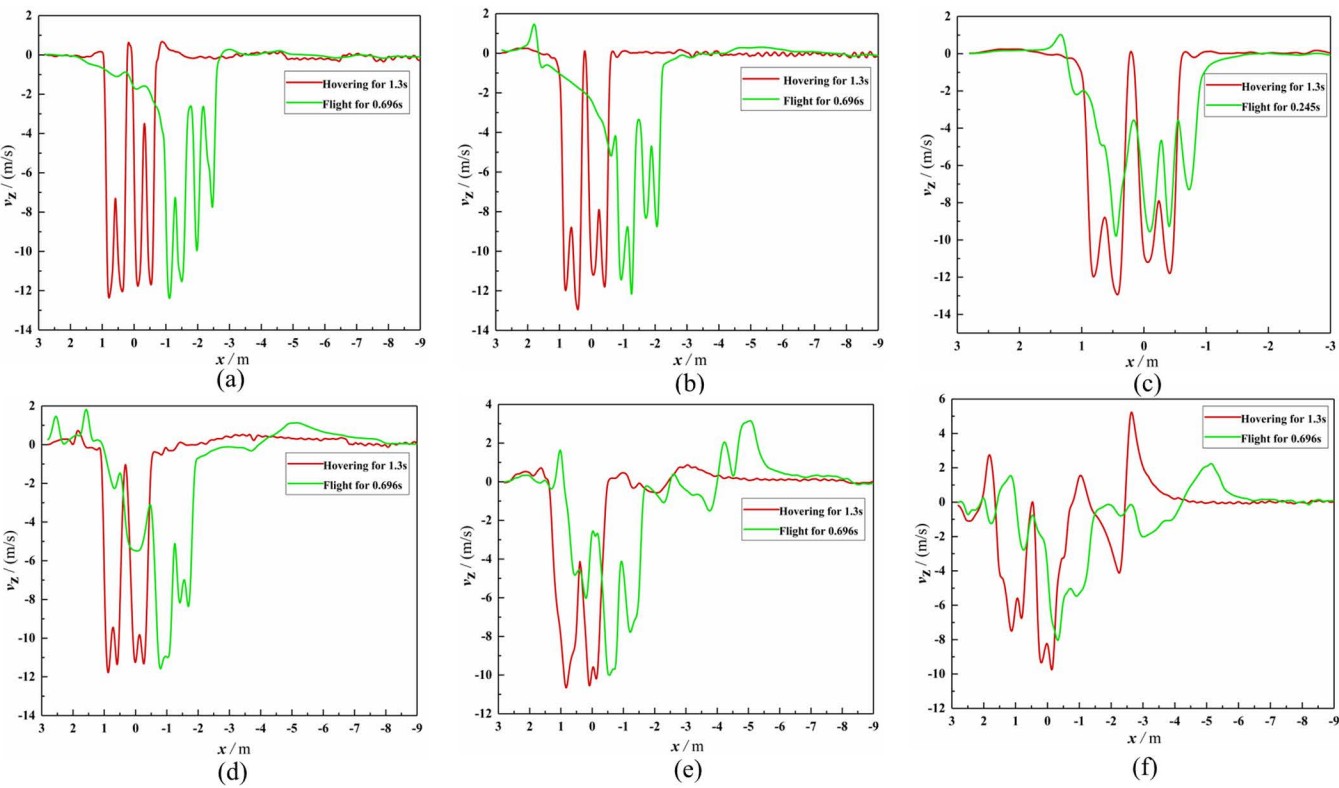

**Fig 11. Comparison chart of $Z_b$-axis velocity on the observation line. (a)** At Line 1 (Working Condition 1). **(b)** At Line 2 (Working Condition 1). **(c)** At Line 2 (Working Condition 1). **(d)** At Line 3 (Working Condition 1). **(e)** At Line 4 (Working Condition 1). **(f)** At Line 5 (Working Condition 1).

body. When the UAV moved in the negative direction of the $X_b$-axis, the air density behind the body would decrease, making the air behind the UAV body flow to the front of the UAV body. Therefore, the airflow in the front direction of the UAV body and the airflow in the rear of the UAV body met at the tail of the UAV body, and formed "vortex areas".

In the analysis of Working Condition 1, it was found that the strong ground effect made Working Condition 1 not suitable for verifying the reliability of numerical calculation. So Working Condition 2 that the UAV flew directly along the $X_b$-axis negative direction at a speed of 3m/s was completed. The velocity distribution diagrams of the cross-section and longitudinal section for Working Condition 2 are shown in Fig 13. The $Z_b$-axis velocity distribution diagrams of Observation Lines 1–5 at different times in Working Condition 2 are also shown in Fig 14. The relative positions of the cross-section, longitudinal section, and the four-rotor plant protection UAV were consistent with Working Condition 1.

By comparing Figs 13a-13c, it could be found that the flight-coupled wind field developed into a dynamically stable state after flying for 1.742 s. It could be seen by combining (Fig 11a, Fig 13c, and Fig 14a) that the dense atmosphere significantly woke the two flight velocity peaks of the front two rotors, and the two flight velocity peaks of the rear two rotors only decreased slightly, which led to two outcomes. The first outcome was that the absolute values of the four $Z_b$-axis velocity peaks from left to right along the $X_b$-axis negative direction at Observation Line 1 decreased from 12.45 m/s, 12.08 m/s, 11.77 m/s, and 11.77 m/s at hover to 11.89 m/s, 11.57 m/s, 10.71 m/s, and 8.70 m/s after the flight was stable, and the attenuation rates of $Z_b$-direction velocity peaks were 4.5%, 4.2%, 9.0%, and 26.1, respectively. The second outcome was that the angles between the horizontal direction and the four curves formed by $Z_b$-axis velocity peaks were 72°, 69°, 61°, and 56°, respectively (as shown in Fig 15), but the angle values were always 90° when hovering. Therefore, to increase the effective deposition amount and improve the spraying effect, it is recommended that the centrifugal nozzle

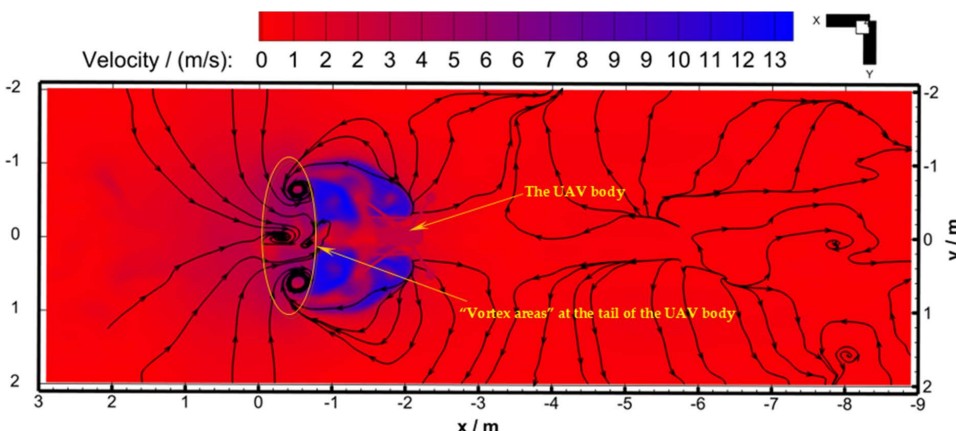

**Fig 12. Horizontal streamline diagram under velocity background at 0.696 s (Working Condition 1).**

should be installed under the rear two rotors, and that the angle between the horizontal direction and the centrifugal nozzle should be set between 18° and 21°. In addition, it can be seen from Fig 14 that the farther the vertical distance away from the UAV rotor, the lower the $Z_b$-axis wind speed value. The flight-coupled wind field would sweep over every point along the flight path, but the closer the vertical distance to the UAV rotor, the more stable the wind speed at the sweep point.

Fig 16 shows the velocity streamline diagram of the longitudinal section (perpendicular to the horizontal direction) of the four-rotor plant protection UAV during flight. Under the pressure induced by rotor rotation, the airflow below and behind the UAV body flowed downward as a whole. The UAV had a forward inclination of 9° and the ground was a non-slip wall boundary. Under the guidance of the flight-coupled wind field that developed diagonally backward, the airflow would eventually develop toward the rear. The flight-coupled wind field that developed diagonally backward had a high speed and could fully develop to the ground, forming a diagonally downward "wind wall". So in the process of flying forward, this "wind wall" will drive the air in front of the UAV body to continue to flow forward. Then, "forward flow areas" were formed in front of and behind the diagonally downward "wind wall".

## 4. Conclusions

This research was motivated by a deep understanding of flight-coupled wind field spatiotemporal evolution law for the four-rotor plant protection UAV. The flight-coupled wind field simulation and the wind speed testing based on the overall UAV were performed, and the conclusions were drawn:

(1) Two simulation working conditions were performed to analyze the evolution of the flight-coupled wind field. The first working condition was that the four-rotor plant protection UAV hovered for 1.3 s, and then flew in the $X_b$-axis negative direction at a speed of 3 m/s. The second working condition was that the four-rotor plant protection UAV flew directly along the $X_b$-axis negative direction at a speed of 3 m/s. Working Condition 1 corresponded to the flow field distribution law study of hover state and flight state switching, Working Condition 2 corresponded to the flight-coupled wind field distribution law study in the flight state. The $Z_b$-axis wind speed values of the observation points in Working Condition 2 were used to compare with the wind speed values of the testing point, and the maximum relative error was 12.7%. The analysis of the flight-coupled wind field based on the numerical calculation was reliable.

(2) In Working Conditions 1 and 2, the evolution of flight-coupled wind field for the four-rotor plant protection UAV had undergone a dynamic process. During the process from hover to flight, the "downwash airflow" experienced lateral interruption, developed back into a dynamically stable state, and eventually became the "flight-coupled wind field"

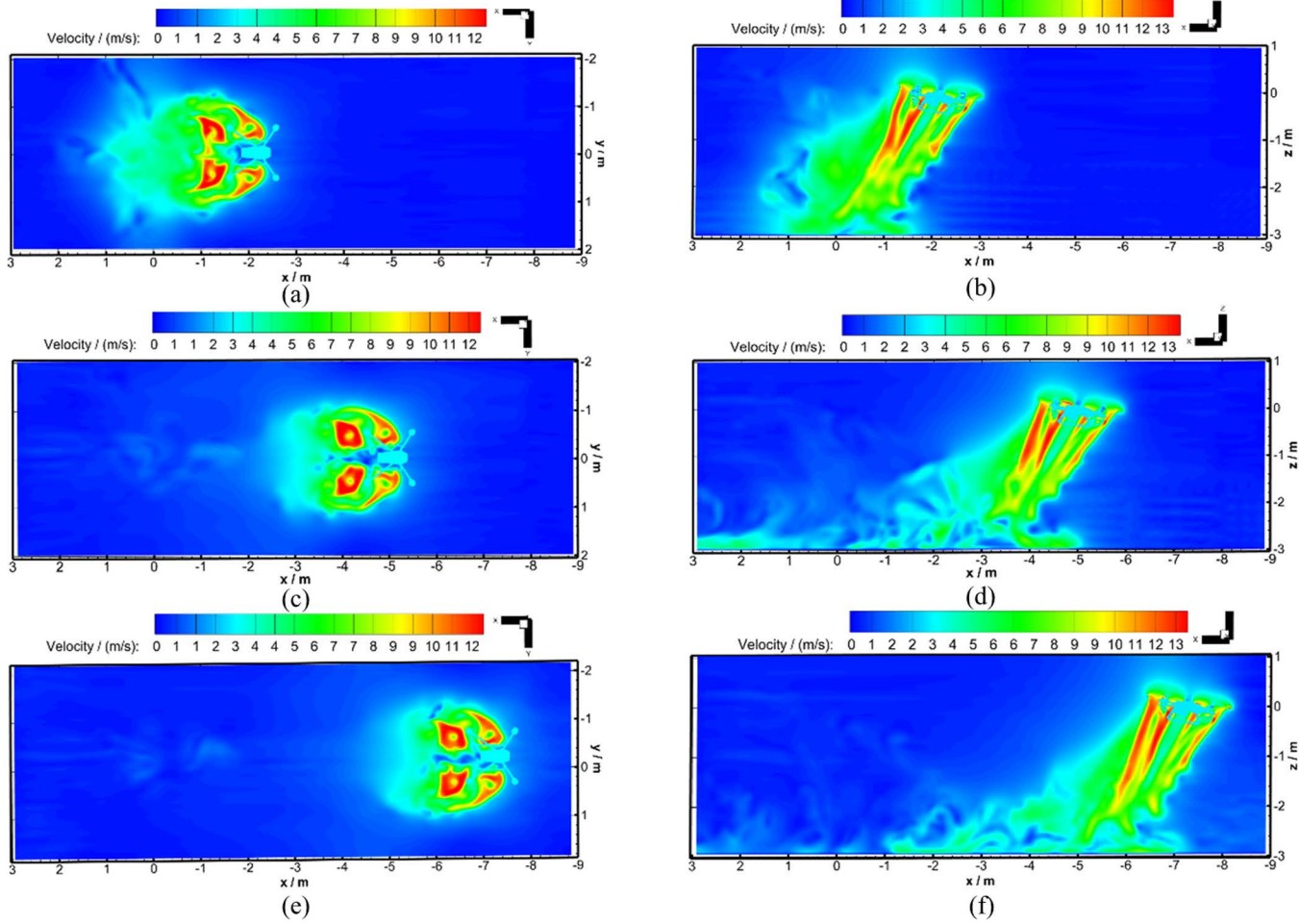

**Fig 13. Horizontal and vertical section velocity diagram at typical moments (No hover process) for the UAV. (a)** After 0.792 s of flight (Working Condition 2). **(b)** After 1.742 s of flight (Working Condition 2). **(c)** After 2.492 s of flight (Working Condition 2).

after flying for 0.696 s. During the process from hover to flight, the "flight-coupled wind field" distribution area of the cross-section developed from directly below the UAV body to gradually behind the UAV body, and finally maintained a stable transverse distance from the UAV body. During the process from hover to flight, the "flight-coupled wind field" was influenced by the incoming airflow from the front, the four circular wind fields on the cross-section developed into four horseshoe tails, which rotated, and spread symmetrically backward across the cross-section in the $X_b$-axis negative direction. During the process from hover to flight, the "flight-coupled wind field" in the longitudinal section developed from relatively stable to the phenomenon that vortices constantly fell off in the opposite direction of flight.

(3) When switching from hover to flight state in Working Condition 1, the "downwash airflow" experienced a lateral interruption and developed into the stable "flight-coupled wind field" after flying for 0.696 s. Therefore, the nozzles were recommended to open and spray until the "flight-coupled wind field" developed into a stable flow, then the massive droplet drift caused by turbulent flow during the initial stage of flight would be significantly reduced.

(4) For the four-rotor plant protection UAV, the dense atmosphere reduced the absolute values of the four $Z_b$-axis velocity peaks from left to right along the flight direction by 4.5%, 4.2%, 9.0%, and 26.1% respectively at Observation Line 1. The

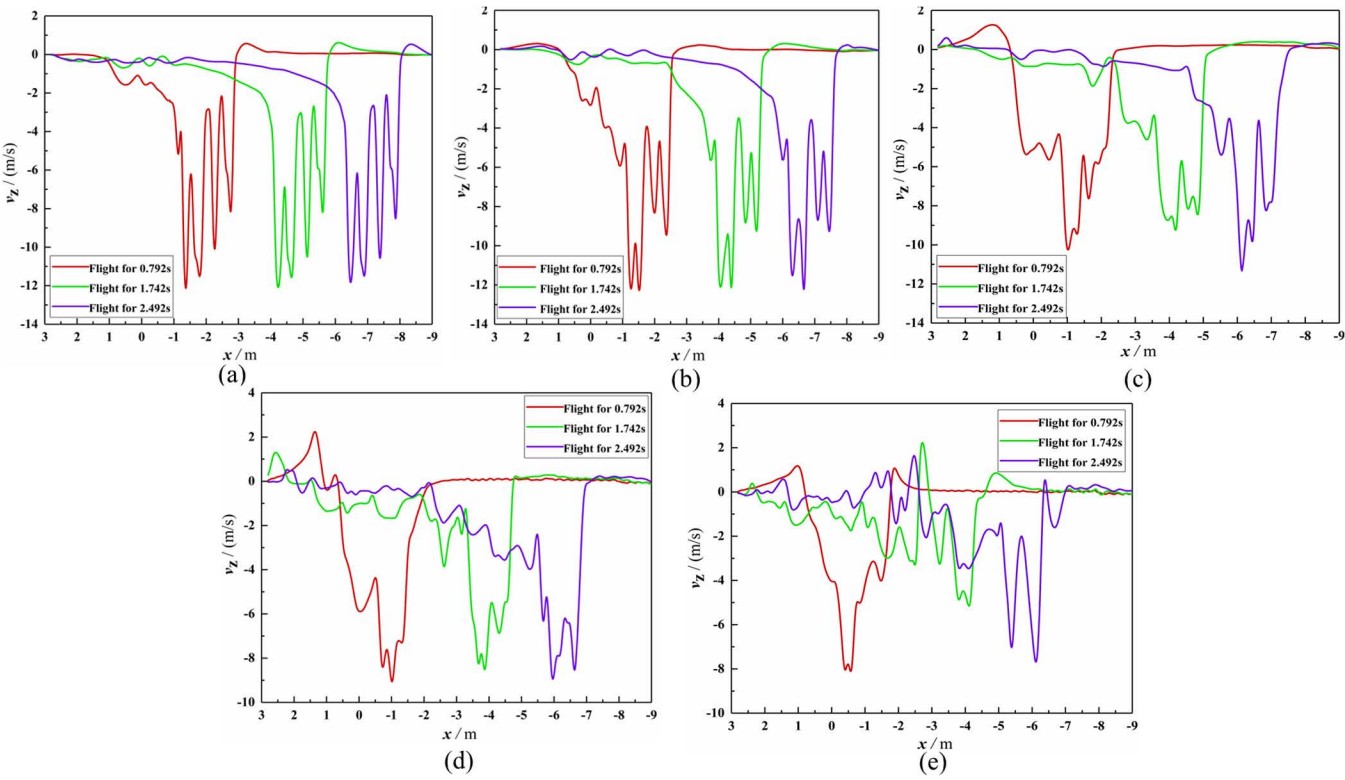

**Fig 14. Comparison chart of $Z_b$-axis velocity on the observation line after flying for some time.** (a) At Line 1 (Working Condition 2). (b) At Line 2 (Working Condition 2). (c) At Line 3 (Working Condition 2). (d) At Line 4 (Working Condition 2). (e) At Line 5 (Working Condition 2).

angles between the horizontal direction and the four curves formed by $Z_b$-axis velocity peaks were also changed from 90° to 72°, 69°, 61°, and 56° respectively. Therefore, to enhance the downward transport effect of the flight-coupled wind field on the droplet groups, and ultimately improve the deposition effect of the canopy, the nozzles were proposed to be installed under the rear two rotors, and the angle between the horizontal direction and centrifugal nozzle (perpendicular to the flight-coupled wind field) should be set between 18° and 21° when the flight speed was 3 m/s.

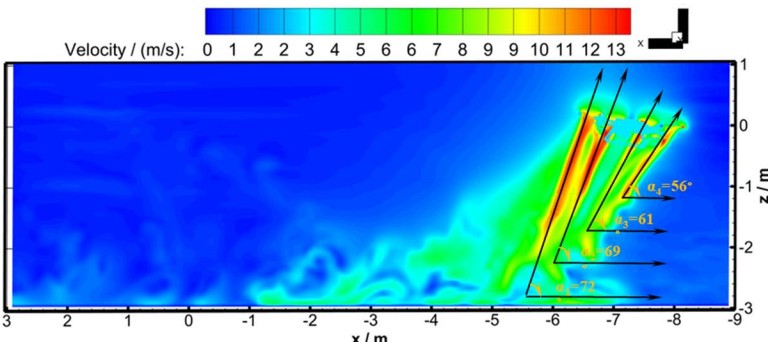

**Fig 15. The angles between horizontality and curves formed by $Z_b$-direction velocity peaks (Working Condition 2).**

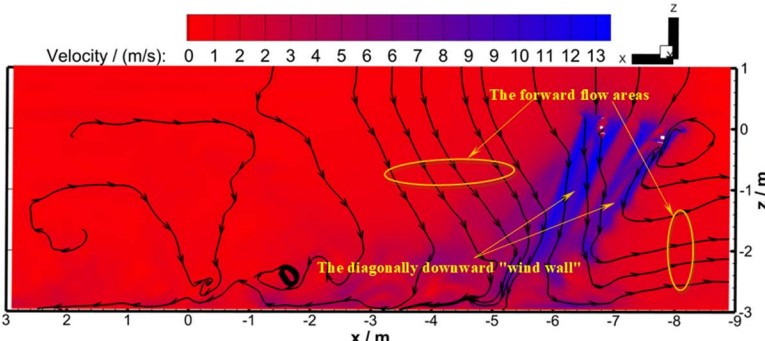

**Fig 16. Longitudinal streamline diagram under velocity background at 2.492 s (Working Condition 2).**

## 5. Future work

The research in this paper can provide a guide for the research on setting the flight buffer distance at different flight speeds before opening the spray nozzle in the future. This work can also lay a foundation for the mechanism study of flight airflow-pesticide droplet-crop canopy interaction for the multi-rotor plant protection UAV.

| Nomenclature | |
|---|---|
| **Abbreviations** | |
| UAV | Unmanned Aerial Vehicle |
| CFD | Computational Fluid Dynamics |
| RANS | Reynolds Averaged Naviere-Stokes |
| RNG | Re-Normalization Group |
| $O_b X_b Y_b Z_b$ | It represents absolute coordinate system |
| $O_e X_e Y_e Z_e$ | It represents relative coordinate system |
| $X_b, Y_b, Z_b$ | $X, Y, Z$ directions in the absolute coordinate system |
| $X_e, Y_e, Z_e$ | $X, Y, Z$ directions in the relative coordinate system |
| **Symbols** | |
| $V$ | Control volume of discrete unit in the computing domain |
| $S$ | Control surface of discrete unit in the computing domain |
| $\vec{W}$ | Conservation variables of the RANS equation |
| $\vec{F}$ | Convective fluxes of the RANS equation |
| $\vec{G}$ | Diffusion fluxes of the RANS equation |
| $\vec{Q}$ | Source fluxes due to the rotation of the rotors |
| $t$ | Current calculation time |
| $H$ | Total enthalpies |
| $E_r$ | Total internal energy |
| $\tau_{xx}, \tau_{yy}, \tau_{zz}, \tau_{xy}, \tau_{xz}, \tau_{yz}; \Phi_x, \Phi_y, \Phi_z$ | Viscous quantities and heat fluxes in three directions respectively |
| $\vec{q}_w$ | Grid velocity perpendicular to the mesh unit surface |
| $\vec{q}$ | Airflow velocity vector |
| $u, \nu, \omega$ | Airflow velocity components in three directions ($x, y, z$) |
| $\rho$ | Airflow density |
| $p$ | Airflow pressure |

**Nomenclature**

| | |
|---|---|
| $\Omega$ | Rotational speed of the rotor |
| $u_i$ | Time mean velocity in the direction $i$ |
| $x_i$ | Coordinate in the $i$ direction |
| $k$ | Turbulent kinetic energy |
| $\varepsilon$ | Turbulent dissipation rate |
| $\mu_{eff}$ | Total viscosity of airflow |
| $\mu_t$ | Turbulent viscosity of airflow |
| $\mu$ | Kinematic viscosity of airflow |
| $G_k$ | The production term of turbulent kinetic energy $k$ due to the average velocity gradient |
| $G_b$ | The production term of turbulent kinetic energy $k$ caused by buoyancy |
| $C_{1\varepsilon}^*$ | An incremental modification of the RNG $k$-$\varepsilon$ equation based on the Standard $k$-$\varepsilon$ equation |
| $\eta$ | The ratio of turbulence to mean flow in time scale |
| $E_{ij}$ | Characteristic strain rate |
| $\alpha_k, \alpha_\varepsilon$ | Turbulence model constants |
| $C_\mu, C_{1\varepsilon}, \eta_0, \beta, C_{2\varepsilon}$ | Constant terms |
| $y+$ | A dimensionless number that characterizes whether the first-layer grid is reasonable |
| $\triangle y$ | The thickness of the first boundary layer |
| $\tau_w$ | The shear stress on the solid wall of the first layer |

## Acknowledgments

The authors are grateful to Nanjing Forestry University for supplying the experimental machines, instruments, workstation used in this work.

## Author contributions

**Data curation:** Zhiwei Li.

**Software:** Fengbo Yang.

**Writing – original draft:** Fengbo Yang.

**Writing – review & editing:** Guangyao Zhang, Hongping Zhou.

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
