## [Decision Letter · Decision Letter 0]

29 Jul 2024

PONE-D-24-27043The Spatiotemporal Evolution of Flight-Coupled Wind Field for a Four-Rotor Plant Protection Unmanned Aerial VehiclePLOS ONE

Dear Dr. Yang,

Thank you for submitting your manuscript to PLOS ONE. After careful consideration, we feel that it has merit but does not fully meet PLOS ONE’s publication criteria as it currently stands. Therefore, we invite you to submit a revised version of the manuscript that addresses the points raised during the review process.

The authors are suggested to provide the procedure for numerical modeling  in detail as suggested by the reviewers.

We look forward to receiving your revised manuscript.

Kind regards,

Muhammad Shakaib, PhD

Academic Editor

PLOS ONE

Journal Requirements:

4. Please note that funding information should not appear in any section or other areas of your manuscript. We will only publish funding information present in the Funding Statement section of the online submission form. Please remove any funding-related text from the manuscript.

    "The authors acknowledge the support provided by the National Natural Science Foundation of China (No. 52275257, No. 51705264)"

6. We note that you have indicated that there are restrictions to data sharing for this study. PLOS only allows data to be available upon request if there are legal or ethical restrictions on sharing data publicly. For more information on unacceptable data access restrictions, please see http://journals.plos.org/plosone/s/data-availability#loc-unacceptable-data-access-restrictions. 

Reviewers' comments:

Reviewer's Responses to Questions

**Comments to the Author**

1. Is the manuscript technically sound, and do the data support the conclusions?

Reviewer #1: Yes

Reviewer #2: No

2. Has the statistical analysis been performed appropriately and rigorously? 

Reviewer #1: Yes

Reviewer #2: N/A

3. Have the authors made all data underlying the findings in their manuscript fully available?

Reviewer #1: Yes

Reviewer #2: No

4. Is the manuscript presented in an intelligible fashion and written in standard English?

Reviewer #1: Yes

Reviewer #2: No

5. Review Comments to the Author

Reviewer #1: 1. Having read the article, I am a little confused on the difference in the terms Working Conditions, Conditions and Calculation Conditions 1 and 2? Do they refer to the same thing?

2.The Equation 1 and 2, I suggest that they come one after the other and not separated by Fig. 2 as the definitions of the symbols are all given in the same paragraph after Equation 2. Fig. 2 should be after the last paragraph of the Introduction section.

3. For Table 1, maybe the columns can be separated into 3 representing x, y and z.

4. For Fig. 5, does the location of the computational domain @ Domain 1 - Domain 4 correspond to the Interface 1-4?

5. is testing point, observation and feature point the same?

6. maybe can explain more on what is the flow field evolution law mentioned.

7. Fig. 6 - the bottom part of the Centrifugal wording seems to be missing.

8. Maybe can use another better wording for the phrase 'dug out' in line 190.

9. For equation 4 , maybe can define each of the symbol used.

10. There are a few suggestions to improve some of the wordings and phrases and also the format of the paper. I have submitted my review of the paper for the author's reference.

Reviewer #2: Overall, the description of the numerical method is insufficient, and there is no reproducibility.

What is the Zb direction? Since there is no explanation, the meaning is unclear within the abstract.

The symbol list should be added.

What grid generation software is used? If the code is in-house, specific schemes should be explained, or published papers should be cited.

What software is being used for the NS solver as well? In the case of in-house code, specific calculation methods

(Space discretization methods (inviscid, viscous), time integration methods, pressure coupling, matrix solvers, etc.) should be explained or published papers should be cited.

The reviewer speculates that some commercial software(StarCCM+ or ANSYS) is used.

I don't know which are the calculation results and which are the experimental results. Are they really compared?

The boundary condition name, such as pressure outlet, does not tell what the boundary condition is(the software's proprietary words).

It is necessary to explain what kind of boundary conditions are imposed on the pressure and velocity by using specific mathematical expressions.

6. PLOS authors have the option to publish the peer review history of their article (what does this mean? ). If published, this will include your full peer review and any attached files.

**Do you want your identity to be public for this peer review?** For information about this choice, including consent withdrawal, please see our Privacy Policy .

Reviewer #1: No

Reviewer #2: No

---

## [Author Response · Author response to Decision Letter 1]

23 Nov 2024

Dear Editor, Dear reviewers,

Thank you for your letter dated July 29th. We thank you very much for the time and effort that they have put into reviewing the previous version of the manuscript. The suggestions have enabled us to improve our work. We would like also to thank you very much for allowing us to resubmit a revised copy of the manuscript. The document (named "Response to Reviewers" in the "Attach Files" section) is our point-by-point response to the comments raised by the reviewers. The comments are reproduced and our responses are given directly afterward in a different color.

---

## [Decision Letter · Decision Letter 1]

5 Jan 2025

PONE-D-24-27043R1The Spatiotemporal Evolution of Flight-Coupled Wind Field for a Four-Rotor Plant Protection Unmanned Aerial VehiclePLOS ONE

Dear Dr. Yang,

Thank you for submitting your manuscript to PLOS ONE. After careful consideration, we feel that it has merit but does not fully meet PLOS ONE’s publication criteria as it currently stands. Therefore, we invite you to submit a revised version of the manuscript that addresses the points raised during the review process.

 The authors are suggested to make minor changes in the paper as recommended by the reviewers

We look forward to receiving your revised manuscript.

Kind regards,

Muhammad Shakaib, PhD

Academic Editor

PLOS ONE

Journal Requirements:

Reviewers' comments:

Reviewer's Responses to Questions

**Comments to the Author**

1. If the authors have adequately addressed your comments raised in a previous round of review and you feel that this manuscript is now acceptable for publication, you may indicate that here to bypass the “Comments to the Author” section, enter your conflict of interest statement in the “Confidential to Editor” section, and submit your "Accept" recommendation.

Reviewer #1: All comments have been addressed

Reviewer #2: All comments have been addressed

2. Is the manuscript technically sound, and do the data support the conclusions?

Reviewer #1: Partly

Reviewer #2: Yes

3. Has the statistical analysis been performed appropriately and rigorously? 

Reviewer #1: N/A

Reviewer #2: N/A

4. Have the authors made all data underlying the findings in their manuscript fully available?

Reviewer #1: (No Response)

Reviewer #2: Yes

5. Is the manuscript presented in an intelligible fashion and written in standard English?

Reviewer #1: Yes

Reviewer #2: No

6. Review Comments to the Author

Reviewer #1: The text needs to be checked for grammatical errors to improve on readers understanding of the information inside the paper. There are still some sentences that require improvement.

Please check again the governing equations and their corresponding explanations. For example. F in the navier-stokes equation may not actual represent inviscid fluxes.

Please check again that all the abbreviations and symbols in the text are defined. For example, y+.

If possible, please use the standard terms usually used in explaining computational fluid dynamics simulation. For example, control body can be replaced by control volume.

Please make sure that the terms used in the text are always consistent. For example, turbulent kinetic energy k and turbulence k equation; epsilon equation and dissipation equation.

It may be useful to make some indications/labels on the results/ figures itself corresponding to the explanation provided in the text. For example, in Figure 10, which is the UV body, airflow, horseshoe tail, etc.....which would really assist the readers in understanding the findings even more.

For Figure 6, please make the figure larger so that the readers may see the mesh/grid more clearly.

Reviewer #2: Thank you for your response to my comment.

There are numbers in the paper with a very large number of significant digits, but it is hard to have such precision in either the experimental or calculated values. All numbers need to be revised to show each with reliable precision.

In Figure 8, the caption states that the calculation results are plotted, but for the other figures, it is not clear whether they draw measurement results or calculation results.

Figure 9 says that it compares experimental and calculated values, but it is not clear which lines are experimental values and which are calculated values.

In several sentences, it is difficult to understand the meaning. For example,

"Only when the coupled wind field is analyzed can the interaction mechanism among wind field, pesticide droplet, and crop canopy be effectively studied, then the flight spray scheme can be reasonably formulated."

in the abstract.

7. PLOS authors have the option to publish the peer review history of their article (what does this mean? ). If published, this will include your full peer review and any attached files.

**Do you want your identity to be public for this peer review?** For information about this choice, including consent withdrawal, please see our Privacy Policy .

Reviewer #1: No

Reviewer #2: No

---

## [Author Response · Author response to Decision Letter 2]

19 Jan 2025

Dear reviewers,

Thank you for your letter dated January 6th. We thank you very much for the time and effort that they have put into reviewing the previous version of the manuscript. The suggestions have enabled us to improve our work. We would like also to thank you very much for allowing us to resubmit a revised copy of the manuscript. The document (named "Response to Reviewers" in the "Attach Files" section) is our point-by-point response to the comments raised by the reviewers. The comments are reproduced and our responses are given directly afterward in a different color.

---

## [Editor Report · Decision Letter 2]

22 Jan 2025

The Spatiotemporal Evolution of Flight-Coupled Wind Field for a Four-Rotor Plant Protection Unmanned Aerial Vehicle

PONE-D-24-27043R2

Dear Dr. Yang,

We’re pleased to inform you that your manuscript has been judged scientifically suitable for publication and will be formally accepted for publication once it meets all outstanding technical requirements.

Kind regards,

Muhammad Shakaib, PhD

Academic Editor

PLOS ONE
---

## [Editor Report · Acceptance letter]

PONE-D-24-27043R2

PLOS ONE

Dear Dr. Yang,

I'm pleased to inform you that your manuscript has been deemed suitable for publication in PLOS ONE. Congratulations! Your manuscript is now being handed over to our production team.

Kind regards,

on behalf of

Dr. Muhammad Shakaib

Academic Editor

PLOS ONE